# Two B-Box Proteins, MaBBX20 and MaBBX51, Coordinate Light-Induced Anthocyanin Biosynthesis in Grape Hyacinth

**DOI:** 10.3390/ijms23105678

**Published:** 2022-05-19

**Authors:** Han Zhang, Jiangyu Wang, Shuting Tian, Wenhui Hao, Lingjuan Du

**Affiliations:** 1College of Landscape Architecture and Arts, Northwest A & F University, Yangling 712100, China; hange@nwafu.edu.cn (H.Z.); wangjiangyu@nwafu.edu.cn (J.W.); tianshuting@nwafu.edu.cn (S.T.); haowenhui123@nwafu.edu.cn (W.H.); 2State Key Laboratory of Crop Stress Biology for Arid Areas, College of Horticulture, Northwest A & F University, Yangling 712100, China; 3Key Laboratory of Horticultural Plant Biology and Germplasm Innovation in Northwest China, Ministry of Agriculture, Yangling 712100, China

**Keywords:** anthocyanin biosynthesis, BBX proteins, flower colour, grape hyacinth, light, molecular mechanism

## Abstract

Floral colour is an important agronomic trait that influences the commercial value of ornamental plants. Anthocyanins are a class of flavonoids and confer diverse colours, and elucidating the molecular mechanisms that regulate their pigmentation could facilitate artificial manipulation of flower colour in ornamental plants. Here, we investigated the regulatory mechanism of light-induced anthocyanin biosynthesis during flower colouration in grape hyacinth (*Muscari* spp.). We studied the function of two B-box proteins, MaBBX20 and MaBBX51. The qPCR revealed that MaBBX20 and MaBBX51 were associated with light-induced anthocyanin biosynthesis. Both MaBBX20 and MaBBX51 are transcript factors and are specifically localised in the nucleus. Besides, overexpression of *MaBBX20* in tobacco slightly increased the anthocyanin content of the petals, but reduced in *MaBBX51* overexpression lines. The yeast one-hybrid assays indicated that MaBBX20 and MaBBX51 did not directly bind to the *MaMybA* or *MaDFR* promoters, but MaHY5 did. The BiFC assay revealed that MaBBX20 and MaBBX51 physically interact with MaHY5. A dual luciferase assay further confirmed that the MaBBX20–MaHY5 complex can strongly activate the *MaMybA* and *MaDFR* transcription in tobacco. Moreover, MaBBX51 hampered MaBBX20–MaHY5 complex formation and repressed *MaMybA* and *MaDFR* transcription by physically interacting with MaHY5 and MaBBX20. Overall, the results suggest that MaBBX20 positively regulates light-induced anthocyanin biosynthesis in grape hyacinth, whereas MaBBX51 is a negative regulator.

## 1. Introduction

Floral colour is an important agronomic trait that determines the commercial value of ornamental plants. Anthocyanins belong to a class of flavonoids and confer a great variety of colours, including pale yellow, red, magenta, violet, and blue to plant flowers and other tissues [1]. Additionally, anthocyanins help attract pollinators, disperse seeds, protect against photo-oxidative damage and provide biotic and abiotic stress tolerance [2,3]. Because of their versatility, elucidating the molecular mechanisms that regulate anthocyanin pigmentation is of considerable importance, especially for the artificial manipulation of flower colour in ornamental plants.

The well-conserved protein complex MYB-basic helix-loop-helix (bHLH)-WDR (MBW) has been demonstrated to control the anthocyanin pathway [4,5]. A core component of MBW is MYB transcription factors (TFs), which primarily determine the spatial and temporal localisation of anthocyanins and combine with common bHLH and WDR factors to activate or inhibit downstream structural genes [4,6,7]. Upstream of MBW complexes, other TFs are also involved in the anthocyanin regulatory network by interacting with MBW complexes to meet the different demands of endogenous developmental signals and external environmental cues [4,5]. Various genetic, ontogenic, morphogenetic, and environmental factors can influence the biosynthesis and accumulation of plant secondary metabolites [8,9]. Environmental stimulations are essential for phenolics compounds (including anthocyanins and other types of flavonoids, but also simple phenols) in plants [10,11]. Among the various environmental stimuli, light is a key factor regulating anthocyanin biosynthesis and accumulation in plants [12,13]. In the light-induced anthocyanin pathway, photoreceptors perceive a visible light signal and then inhibit the ubiquitin E3 ligase activity of constitutively photomorphogenic-1 (COP1), which mediates the ubiquitination and degradation of the light-response effector, Elongated Hypocotyl-5 (HY5) [14]. HY5, a key TF in light-regulated anthocyanin biosynthesis, directly binds to the promoters of *chalcone synthase* (*CHS*), *chalcone isomerase* (*CH**I*), and *production of anthocyanin pigment 1* (*PAP1*) to control their expression and pigment accumulation [15,16,17]. In addition to HY5, R2R3 MYBs are one of the most important TFs that are involved in light-induced anthocyanin biosynthesis and are also degraded by COP1, the molecular switch in the light signalling pathway. Many light-inducible R2R3 MYBs that control anthocyanin biosynthesis have been identified in numerous plants [12]. For example, high light-induced expression of the regulator *FvMYB10* in flower petals of *Fragaria vesca*; furthermore, *FvMYB10* in strawberry is a light-inducible R2R3 MYB involved in anthocyanin accumulation regulation [18]. In addition, high light intensity treatment can up-regulate the expression of *MdMYB1* in apples, leading to higher expression of structural genes in the colour-forming pathway, and ultimately promoting anthocyanin accumulation in the apple peel [19]. In addition, increasing evidence indicates that some B-box (BBX) proteins, in association with HY5, play important roles in the regulation of anthocyanin accumulation in response to light in plants [20].

BBX proteins belong to a subclass of zinc finger TFs that contain one or two B-box domains in the N terminus, which are involved in protein–protein interactions, nuclear protein transport, and transcriptional regulation [21,22]. Some BBX proteins also possess a CONSTANS, CO-like, and TOC1 (CCT) domain in their C terminus [23]. Based on the number and type of domains carried, BBX proteins can be divided into five structural groups [24]. Two types of BBX regulators involved in the fine-tuning of light-regulated anthocyanin pigmentation, i.e., positive and negative BBX regulatory proteins, mainly belong to structural group IV or V [25]. In *Arabidopsis*, AtBBX21 positively regulates anthocyanin accumulation and inhibits the elongation of the seedling hypocotyl in response to light by binding to the promoters of *chalcone isomerase* and *HY5* and directly activating their expression, both independently and together with HY5 [26]. Other BBX proteins play similar positive regulatory roles in the light-induced anthocyanin pathway, such as AtBBX20 [27], AtBBX21 [26,28], AtBBX22 [29,30], and AtBBX23 [31]. Conversely, AtBBX24 and AtBBX25 negatively regulate anthocyanin accumulation by interacting with HY5 and inhibiting *AtBBX22* expression [32]. AtBBX32 negatively modulates anthocyanin accumulation by interacting with AtBBX21 to suppress AtBBX21-HY5 activity [33]. Recent studies have demonstrated that BBX proteins are also involved in regulating anthocyanin accumulation during fruit skin colouration [20,25,34,35,36]. In apple plants, MdCOL11 [37], MdBBX20 [38], and MdBBX22 [39] are positive regulators of UV-B-induced anthocyanin biosynthesis, whereas MdCOL4 [36] and MdBBX37 [40] are negative regulators in response to UV-B/light radiation of fruit skin. In the pear plants, PpBBX16 [34] and PpBBX18 [35] have been identified as positive regulators of light-induced anthocyanin biosynthesis. Moreover, PpBBX21 represses light-induced anthocyanin biosynthesis by interacting with PpHY5 and PpBBX18 to inhibit the formation of the PpHY5-PpBBX18 complex [35]. A 14-nucleotide deletion mutation in the coding region of the PpBBX24 gene may be a key factor that causes the red colour of ‘Zaosu Red’ pears [25]. A recent study has demonstrated that SlBBX20 interacts with SlCSN5-2, a COP9 signalosome subunit, and promotes anthocyanin biosynthesis by binding the dihydroflavonol reductase promoter in tomatoes [41]. However, the regulatory mechanisms underlying anthocyanin biosynthesis mediated by BBX family members during flower colouration of ornamental plants are not fully understood.

Grape hyacinth (*Muscari* spp.) is a popular bulbous floricultural crop used as a garden and indoor potted plant because of its brilliant flower colours, especially the blue colouration, which is considered mysterious and romantic [42,43,44]. It is often used as a model plant for studying the molecular mechanisms underlying anthocyanin biosynthesis in monocotyledonous plants [42,45]. Our previous studies have shown that two R2R3-MYBs (MaAN2 and MaMybA) and one R3-MYB (MaMYBx) function as transcriptional activators and an inhibitor, respectively, to coordinate anthocyanin biosynthesis with the cofactor MabHLH1 in grape hyacinth [43,44,45]. Moreover, our cloned bZIP TF MaHY5 [46] was found to activate the *GUS* reporter gene driven by *MaMybA* promoters in a *GUS* reporter system (data not published). Thus, MaHY5 may be associated with anthocyanin biosynthesis in grape hyacinth. However, the transcriptional regulatory mechanisms located upstream of MYBs during anthocyanin biosynthesis in grape hyacinth are yet to be elucidated.

Here, we studied the function of two BBX proteins, MaBBX20 and MaBBX51. The shading treatment and heterologous overexpression tobacco assays revealed that they positively and negatively regulate light-induced anthocyanin biosynthesis during flower pigmentation, respectively. This study provides insights into the transcriptional regulatory mechanisms upstream of MYBs, which emphasizes MaBBX20 and MaBBX51 as new targets for the genetic manipulation of grape hyacinth flower colour.

## 2. Results

### 2.1. MaBBX20 and MaBBX51 Belong to the BBX Protein Family

BBX TFs play a key role in the synthesis of anthocyanin. In this study, combining the transcriptome-related data [47] of the grape hyacinth flower in the early stage, through the homologous protein local BLASTP and gene function annotation, we obtained two BBX unigenes related to grape hyacinth. The two BBXs unigenes related to flower colour were cloned from *M. aucheri* ‘Dark Eyes’ flowers and named MaBBX20 and MaBBX51. The open reading frames (ORFs) of MaBBX20 and MaBBX51 were 1245 bp and 723 bp in length, encoding proteins with 414 and 240 amino acid residues, respectively (GenBank accession numbers: MW160173 and MW160172). Multiple sequence alignment revealed that the amino acid sequences of MaBBX20 and MaBBX51 contained two tandem BBX domains at the N-terminus, whereas MaBBX20 contained an extra CCT domain at the C-terminus (Figure 1).

To further explore the evolutionary relationships between the two MaBBXs and the other BBX proteins, we constructed a phylogenetic tree of the two MaBBXs and 32 BBX proteins in *Arabidopsis* (Figure 2). MaBBX20 was clustered into subfamily II and shared high sequence similarity with AtBBX7 and AtBBX8 [48] (Figure 2), whereas MaBBX51 belonged to subfamily IV and showed the highest similarity with AtBBX24 and AtBBX25, which are closely related to anthocyanin synthesis [25,32] (Figure 2).

### 2.2. MaBBX20 and MaBBX51 Are Transcriptional Activators Localised in the Nuclei

To determine the subcellular localisation of the two MaBBXs, N. benthamiana leaves were separately transfected with *35S:GFP*, *35S:MaBBX20:GFP*, and *35S:MaBBX51:GFP* fusion constructs. As shown in Figure 3A, the green fluorescence signal of the control *35S:GFP* was not specifically localised in the nucleus. Whereas that of *35S:MaBBX20:GFP* and *35S:MaBBX51:GFP* were specifically localised in the nuclei, indicating that MaBBX20 and MaBBX51 are nuclear localised.

To determine whether MaBBX20 and MaBBX51 have transcriptional activation ability, we performed a transactivation assay in yeast. As shown in Figure 3B, yeast transformed with the positive control (pGBKT7-p53 + pGADT7-T), pGBKT7/MaBBX20 and pGBKT7/MaBBX51 plasmids were able to grow in SD/-Trp media (supplemented with 40 mg mL-1 X-α-gal and 200 ng mL-1 AbA) and exhibited blue plaques, whereas the negative control (pGBKT7 alone) did not grow under the same conditions. The results indicate that MaBBX20 and MaBBX51 show trans-activation activity in the yeast system. Taken together, the results indicate that MaBBX20 and MaBBX51 function as TFs.

### 2.3. Flower Anthocyanin Levels and Expression Analysis of MaBBX20 and MaBBX51 under Shade Treatment

There are numerous reports on the role of BBX proteins in fine-tuning the regulation of light-induced anthocyanin accumulation [28,34,35]. Thus, we sought to characterise the regulation of *MaBBX20* and *MaBBX51* expression; to this end, we isolated the 1313 bp and 1056 bp promoter fragments of *MaBBX20* (*pMaBBX20*; MW561193) and *MaBBX51* (*pMaBBX51*; MW561192), respectively, from *M. aucheri* ‘Dark Eyes’. Analysis of putative regulatory elements using the PlantCARE database (Appendix A) revealed that the promoters contained predicted light-related elements, such as the GT1 (5′-GTGTGTGAA-3′), TCCC (5′-TCTCCCT-3′), GATA (5′-GATAGGG-3′) and TCT (5′-TCTTAC-3′) motifs and an AE-box (5′-AGAAACAA-3′). The presence of potential light-related elements in the upstream region of *MaBBX20* and *MaBBX51* suggests that these two genes are regulated by light.

First, we detected differences in the expression of *MaBBX51* and *MaBBX20* in various tissues (roots, bulbs, leaves, old leaves, and flowers) of *M. aucheri* ‘Dark Eyes’. The qPCR analysis results showed that *MaBBX51* was relatively highly expressed in young leaves, old leaves, and flowers, and that *MaBBX20* was relatively highly expressed in old leaves and flowers (Appendix A). To further evaluate whether the expression of the two MaBBX genes is correlated with light-induced anthocyanin accumulation in grape hyacinth, we used aluminium foil to cover the whole inflorescence during the floral developmental stages (S1–S5) of *M. aucheri* ‘Dark Eyes’. As shown in Figure 4A, the shade condition blocked anthocyanin accumulation in the flowers, resulting in etiolated phenotypes; high-performance liquid chromatography (HPLC) also confirmed that the total anthocyanin levels of the whole inflorescence were significantly reduced, indicating that the flower colouration depends on light exposure (Figure 4B). Moreover, the expression levels of *MaBBX20* and *MaBBX51* were downregulated to different degrees during S2–S5 after shade treatment, similar to the downregulation of the anthocyanin-related genes *MaMybA* and *MaDFR* (Figure 4C). The light-sensitive stage of MaBBX51 was S2, whereas that of MaBBX20 was S3–S5. MaHY5 also showed downregulation after shade treatment during S2–S4, especially in S3. Taken together, the above results indicate that the two *MaBBX* genes and *MaHY5* may be light-induced TFs involved in the regulation of anthocyanin synthesis in grape hyacinth.

### 2.4. The Two BBX Regulatory Genes Are Related to Anthocyanin Biosynthesis

To verify the functions of *MaBBX20* and *MaBBX51*, stable transgenic tobacco (*N. tabacum* ‘SR-1′) plants were generated to heterologous expressions *MaBBX20* and *MaBBX51* driven by the cauliflower mosaic virus 35S (*CaMV35S*) promoter. Eight independent *MaBBX20* overexpression (OE) lines were generated, none of which showed significant phenotypic changes of anthocyanin accumulation that can be evaluated by the naked eye (Figure 5A). Eleven *MaBBX51* OE lines were generated, of which seven had a reduced anthocyanin phenotype (Figure 5A). Compared with the WT control line, the total anthocyanin levels of *MaBBX20* OE lines were slightly increased, whereas those of MaBBX51 OE lines were significantly reduced (Figure 5B). qPCR revealed that the expression levels of the anthocyanin biosynthetic structural genes *NtCHS*, *NtF3H*, *NtDFR*, *NtANS*, and *NtUFGT* in *MaBBX20* OE lines were upregulated, whereas those of *NtCHS*, *NtDFR*, *NtANS*, and *NtUFGT* in *MaBBX51* OE lines were downregulated (Figure 5C–E). Taken together, these results indicate that MaBBX20 and MaBBX51 may regulate anthocyanin biosynthesis in tobacco plants and perform positive and negative regulatory functions, respectively.

### 2.5. The Two BBX Proteins Interact with MaHY5 and Jointly Regulate Anthocyanin Biosynthesis in Grape Hyacinth

Because many BBXs have been characterised as regulating anthocyanin biosynthesis in an HY5-dependent manner in other plants, we performed bimolecular fluorescence complementation (BiFC) to determine whether the two MaBBXs interact with MaHY5 in vivo. As shown in Figure 6., YFP fluorescence was detected in the nuclei of *N. benthamiana* leaf epidermal cells when pSPYCE/MaHY5 + pSPYNE/MaBBX20 and pSPYNE/MaHY5 + pSPYCE/MaBBX51 were transformed into the protoplast, whereas no YFP fluorescence was observed in any of the negative control groups. The results indicate that MaBBX20 and MaBBX51 physically interact with MaHY5. Additionally, we observed physical interactions between MaBBX20 and MaBBX51 (Figure 6B). We next examined whether the two MaBBXs physically interact with the key anthocyanin regulator MaMybA in grape hyacinth [44], given that MdBBX37 negatively regulates anthocyanin biosynthesis by directly interacting with MdMYB1 and MdMYB9, inhibiting their binding to target genes [40]. We found that MaBBX51, but not MaBBX20, interacted with MaMybA (Figure 6A,B). To ascertain how MaBBX20 and MaBBX51 regulate light-induced anthocyanin biosynthesis in grape hyacinth, the correlations between the two BBX proteins and anthocyanin-related genes were analysed. Firstly, direct interactions between MaBBX20, MaBBX51, and the promoters of anthocyanin-related genes *MaMybA* or *MaDFR* were determined using the yeast one-hybrid assay (Figure 7A). MaBBX20 and MaBBX51 did not bind to the promoter regions of either *MaMybA* or *MaDFR* (Figure 7A), although these regions contained putative G–box elements (Figure 7B). However, MaHY5 was able to interact with the promoters of *MaMybA* and *MaDFR* (Figure 7A).

We further analysed the trans-activation capability of MaBBX20, MaHY5, and MaBBX51 alone or in a complex on *pMaMybA* and *pMaDFR* in *N. benthamiana* leaves using a dual-luciferase reporter assay. Infiltration of MaBBX20 alone activated *pMaMybA* and *pMaDFR* transcription, whereas infiltration of MaHY5 alone slightly activated transcription (Figure 7C). Co-infiltration of MaBBX20 and MaHY5 strongly activated *pMaMybA* and *pMaDFR* transcription, indicating that the MaBBX20–MaHY5 complex can induce the transcription of *MaMybA* or *MaDFR* (Figure 7C).

Given that MaBBX51 also physically interacted with MaBBX20 and MaHY5 (Figure 6B), we further analysed the interaction of these proteins together. As expected, infiltration of MaBBX51 alone or with MaHY5 did not activate *pMaMybA* and *pMaDFR* transcription (Figure 7C). Moreover, the activation activities of MaBBX20 alone or in a complex with MaHY5 complex were suppressed upon co-infiltration with MaBBX51 (Figure 7C). These results indicate that the MaBBX51-MaHY5 complex cannot induce transcription of anthocyanin-related genes and that MaBBX51 disturbs MaBBX20 alone or the MaBBX20–MaHY5 complex, inhibiting the transcription of anthocyanin-related genes in grape hyacinth.

We further analysed the interaction between MaBBX51 and MaMybA, as they were found to physically interact (Figure 6B). Similar to our previous results [44,45], infiltration of MaMybA alone as a control activated the promoters of both itself and MaDFR (Figure 7D). However, when MaMybA and MaBBX51 were co-infiltrated, *pMaMybA* and *pMaDFR* promoter activation was repressed (Figure 7D). These results reveal that MaBBX51 negatively regulates anthocyanin biosynthesis by interfering with the binding of MaMybA to its target genes.

## 3. Discussions

Light, an important environmental factor, can control anthocyanin accumulation in plant tissues, especially in petals and fruits [49]. In grape hyacinth, light regulated the biosynthesis of anthocyanin in flowers and multiple light-related elements, such as the G-box elements, which were found in the upstream region of the key anthocyanin activator *MaMybA* and its target gene *MaDFR* (Figure 7B) [44], confirming that light is an important environmental factor for MYB-mediated anthocyanin biosynthesis. However, the molecular mechanisms that occur upstream of *MaMybA* remain elusive. In the present study, we characterised two BBX proteins, MaBBX20 and MaBBX51, and found that they interact with MaHY5 to coordinate the regulation of light-induced anthocyanin biosynthesis in grape hyacinth.

Several studies have shown that BBX group IV proteins containing two tandem repeat B-box domains, but no CCT domain, can act as either positive or negative regulators of light-induced anthocyanin biosynthesis in plants [24,36,50,51]. In addition, some BBX group V proteins contain only one B-box domain, such as AtBBX32 in *Arabidopsis* [33] and MdBBX37 in apple [40], and are involved in negatively controlling anthocyanin accumulation. In the present study, we identified a novel BBX protein MaBBX20 belonging to group II that shared high sequence similarity with AtBBX7 (COL9) from *Arabidopsis*. *Arabidopsis BBX7* possesses two B-box domains and one CCT domain and predominantly functions as a negative regulator of flowering in the photoperiod pathway [48]. CmBBX8, a homolog of AtBBX7, is instead a floral activator in the chrysanthemum photoperiod pathway [52]. In the present study, we revealed the role of MaBBX20 as a positive regulator in light-induced anthocyanin biosynthesis in grape hyacinth. Additionally, another BBX protein MaBBX51, a homolog of AtBBX24, belonging to group IV was identified. In Arabidopsis, AtBBX24 represses anthocyanin accumulation by heterodimerising with HY5 to inhibit binding with DNA [28]. In addition to AtBBX24, orthologs in other plants, such as MdCOL4 in apple [36] and PpBBX24 in pear [25], show similar functions. Given the functional conservation of BBXs, we speculated that MaBBX51 may negatively regulate anthocyanin biosynthesis.

Throughout flower development, the expression pattern of *MaBBX51* was most consistent with that of the anthocyanin-related genes *MaMybA* and *MaDFR*, whose expression is closely related to anthocyanin accumulation in ‘Dark Eyes’, indicating that MaBBX51 is primarily implicated in anthocyanin biosynthesis. However, MaBBX20 and MaHY5 were not closely correlated with anthocyanin accumulation. We speculated that MaBBX20 and MaHY5 may play pleiotropic roles in various developmental processes, in addition to anthocyanin biosynthesis, in grape hyacinth. Moreover, *MaBBX20* and *MaBBX51* overexpression showed slightly enhanced and repressed anthocyanin accumulation as well as minor to noticeable phenotypic changes, respectively. Thus, MaBBX20 may be partially and positively involved in anthocyanin biosynthesis, whereas MaBBX51 has a clear role in suppressing anthocyanin biosynthesis. In addition, based on phenotypic observation, there were no significant differences in the colours of leaves, petals, calyx, filaments, and other tissues between MaHY5 transgenic-lines and WT-lines (data not shown). The results showed that MaHY5 may not facilitate adequate anthocyanin accumulation in transgenic tobacco flowers, indicating that HY5 needs to interact with other proteins (such as BBXs) to regulate anthocyanin biosynthesis.

As a central factor in the light signal transduction pathway, HY5 can integrate light signals with anthocyanin accumulation by directly binding to the promoters of anthocyanin-related genes in *Arabidopsis*, apple, and pear [17,50,53,54]. BBX group IV proteins have been described as HY5-interacting factors that largely depend on HY5 for their function to mediate anthocyanin accumulation in light conditions [26,28,30,31,34,35,36,38,40]. In the present study, we confirmed that MaBBX20 and MaBBX51 physically interact with MaHY5 in vivo (Figure 6), suggesting that HY5-interacting BBX proteins do not exclusively belong to group IV and that other BBX subgroup proteins may be implicated in light-induced anthocyanin biosynthesis. The results of the yeast one-hybrid assay indicate that MaBBX20 and MaBBX51 mediate anthocyanin biosynthesis in an HY5-dependent manner, which is similar to that of PpBBX16, PpBBX18, and PpBBX21 in pear [34,35]. Moreover, given that MaBBX20 did not directly bind to *pMaMybA* or *pMaDFR* but possessed trans-acting activity (Figure 3B), we speculated that MaBBX20 activates *pMaMybA* or *pMaDFR* transcription by interacting with endogenous HY5 in tobacco plants. Thus, these results confirmed that MaBBX20 is dependent on HY5. In *Arabidopsis*, AtBBX24 represses anthocyanin accumulation by inhibiting HY5 binding to the promoter of anthocyanin-related genes [28]. Pear PpBBX21 represses anthocyanin biosynthesis by hampering the formation of the PpHY5–PpBBX18 complex by interacting with PpHY5 and PpBBX18 [35]. As their homologs, co-infiltration of MaBBX51 with MaBBX20 or the MaBBX20–MaHY5 complex suppressed transcription activation (Figure 7C). Given that MaBBX51 can interact with MaBBX20 and MaHY5 (Figure 6B), MaBBX51 should interfere with the binding of MaHY5 to the *MaMybA* and *MaDFR* promoters, possibly by heterodimerising with MaHY5 and homodimerising with MaBBX20. MaBBX51 also interacted with MaMybA (Figure 6B), which is known to positively regulate anthocyanin biosynthesis in grape hyacinth [44,45]. Thus, MaBBX51 may be involved in negatively regulating the anthocyanin biosynthesis through other pathways, e.g., by directly interfering with *MaMybA* binding to its target genes such as *MaDFR*. Indeed, this was confirmed when MaBBX51 and MaMybA were co-infiltrated in the dual-luciferase assay (Figure 7D). Therefore, the results indicate that BBX proteins can interact with various other factors to form homologous or heterologous dimers to fine-tune the regulation of plant growth and development [20,22,35,40,55].

Overall, our findings indicate that MaBBX20, MaBBX51, and MaHY5 mediate light-induced anthocyanin biosynthesis in grape hyacinth (Figure 8). Under light conditions, (1) MaBBX20 physically interacts with MaHY5 and forms a MaBBX20–MaHY5 complex to activate *MaMybA* and *MaDFR* transcription; (2) MaBBX51 physically interacts with MaHY5 and MaBBX20, hampering the formation of the MaBBX20–MaHY5 complex and subsequently repressing *MaMybA* and *MaDFR* transcription and (3) MaBBX51 interfere with *MaMybA*, binding to its promoter and that of *MaDFR*, inhibiting *MaMybA* and *MaDFR* expression. Notably, MaBBX20, a member of BBX group II, may only be partially involved in the light-induced anthocyanin regulatory network in grape hyacinth. Thus, there should be other BBX proteins functioning in this network. Future research should focus on identifying more BBX members to elucidate the sophisticated regulatory mechanism that occurs upstream of *MaMybA* during light-induced anthocyanin biosynthesis in grape hyacinth. Nevertheless, the results provide novel insights into this upstream network and our findings will be useful for designing strategies to modify flower colour in ornamental plants by controlling light conditions.

## 4. Materials and Methods

### 4.1. Plant Materials and Growth Conditions

Grape hyacinth cultivars (*Muscari aucheri* ‘Dark Eyes’) in the fields of Northwest A&F University (Yangling District, Shaanxi Province, China) were examined. Floral development was divided into five stages (S1–S5) as described by Zhang et al., (2020) [45]: S1, closed buds, no pigmentation; S2, closed buds, tepal pigmentation starts; S3, closed buds just before blooming; S4, opening of flowers and S5, flowers during senescence [44]. For flower shade treatment, the entire inflorescences were covered with aluminium foil when the inflorescences sprouted out of the bulbs. The shade covering was large enough to allow for inflorescence growth. The other uncovered inflorescences were exposed to natural light conditions as the control treatment. Fresh flowers from the shade and control treatments were immediately frozen in liquid nitrogen and stored at −80 °C for total anthocyanin measurement and gene expression analysis. Aseptic tobacco (*Nicotiana tabacum* ‘SR-1’) seedlings were used for transformation experiments as described by Chen et al. (2019) [44]. Wild-type and T1 generation transgenic plants were grown in a containment greenhouse (16 h photoperiod, 23 ± 2 °C/17 ± 2 °C day/night temperature). The fully opened flowers were harvested and stored at −80 °C until further analysis. *Arabidopsis thaliana* and *Nicotiana benthamiana* plants were grown in a light incubator (16 h photoperiod, 22 °C) until reaching the four to six-leaf stage for the bimolecular fluorescence complementation (BiFC) assay, subcellular location, and dual-luciferase assays, respectively.

### 4.2. Gene Isolation, Sequence Alignment, and Phylogenetic Analysis

Two BBX unigenes were obtained from the transcriptome of grape hyacinth [47] and designated MaBBX20 and MaBBX51. The ORF sequences of MaBBX20 and MaBBX51 were amplified from the cDNA of *M. aucheri* ‘Dark Eyes’, using the primer pairs listed in Appendix A. MaBBX20 and MaBBX51 cDNA was submitted to the NCBI GenBank database (accession numbers: MW160173 and MW160172). Multi-sequence alignment was performed using the GenDoc software. The phylogenetic tree of MaBBX20, MaBBX51, and 32 BBX proteins of *A. thaliana* [24] were constructed using the MEGAX software with the neighbour-joining algorithm and 1000 bootstrap replicates. The accession numbers of all protein sequences are listed in Appendix A.

### 4.3. RNA Isolation and qPCR

RNA extraction, reverse transcription and quantitative real-time polymerase chain reaction (qPCR) were conducted following the manufacturers’ protocols as previously described by Chen et al. (2017) [43]. The qRT-PCR primers of MaBBXs were designed using Oligo Calc (http://biotools.nubic.northwestern.edu/OligoCalc.html, accessed on 5 March 2022). All primers for qPCR are listed in Appendix A. *MaActin* and *NtTubA1* were used as internal controls for grape hyacinth and tobacco gene expression, respectively. The expression levels of each gene were calculated using the method described by Schmittgen and Livak (2007) [56]. Analysis was performed using three biological and technical replicates.

### 4.4. Tobacco Plant Transformation

Two overexpression vectors (*35S:MaBBX20*: green fluorescent protein (GFP) and *35S:MaBBX51:GFP*) were constructed from binary vectors harbouring a pCAMBIA2300-GFP (*35S:GFP*) backbone as previously described by Zhang et al. (2020) [45]. Then, they were transformed into *Agrobacterium tumefaciens* GV3101 cells using the freeze-thaw method. *Agrobacterium tumefaciens*-mediated transformation was performed to generate transgenic tobacco (*N. tabacum* ‘SR-1′) plants. The homozygous T1 *MaBBX20* overexpression lines (2#, 3# and 6#) and *MaBBX51* overexpression lines (4#, 12# and 15#) that exhibited phenotypic changes in flower pigmentation were used for further high-performance liquid chromatography (HPLC) and qPCR analyses.

### 4.5. Subcellular Localisation and Transcription Activation Assays

For subcellular localisation assays, the *35S:MaBBX20:GFP* and *35S:MaBBX51:GFP* constructs were introduced into tobacco (*N. benthamiana*) epidermal cells via *Agrobacterium*-infiltrated tobacco leaves. Samples transformed with *35S:GFP* were used as control. After 16–30 h of cultivation, transformed materials were monitored for GFP activity under a confocal laser-scanning microscope (TCS SP8; Leica, Wetzlar, Germany).

To measure the transcription activation ability of MaBBX20 and MaBBX51, the full-length cDNAs of MaBBX20 and MaBBX51 were separately cloned into the pGBKT7 vector; the primers used are listed in Appendix A. The constructed vectors, together with the positive (pGBKT7-p53 + pGADT7-T) and negative (pGADT7) controls, were independently transformed into yeast strain Y2HGold. Yeast transformation and auto-activation testing were performed as previously described [43].

### 4.6. BiFC Assay

For BiFC assays, *MaBBX51*, *MaBBX20*, *MaHY5*, and *MaMybA* without termination codons were recombined into the pSPYCE (M) vector, whereas those with termination codons were recombined into the pSPYNE (R)173 vector [44,45]; the primers used are listed in Appendix A. The recombinant and/or empty vectors were transformed into *A. thaliana* mesophyll protoplasts using polyethylene glycol [57]. After transformation for 20 h, the YFP signal was imaged using the TCS SP8 confocal laser microscope.

### 4.7. Yeast One-Hybrid Assay

Yeast one-hybrid assays were performed using the Matchmaker Gold Yeast One-Hybrid System Kit (TaKaRa Bio, Shiga, Japan) according to the manufacturer’s instructions. In brief, the ORF sequences of *MaBBX51*, *MaBBX20*, and *MaHY5* were amplified and inserted into pGADT7 to generate the prey vectors. The promoter sequences of *MaMybA* and *MaDFR* were amplified and inserted into the pAbAi vector to generate the bait vectors. The primers used for this assay are listed in Appendix A. The pAbAi bait vectors were linearised and transformed into yeast strain Y1HGold, after which the prey vectors were transformed into Y1HGold cells harbouring the pAbAi-bait and tested on SD/-Leu/AbA plates. Yeast transformed with the positive and negative controls, pGBKT7/MaBBX20 and pGBKT7/MaBBX51 vectors, were cultivated in SD/-Trp medium and SD/-Trp medium with 40 mg mL^−1^ X-α-Gal and SD/-Trp medium plus 40 mg mL^−1^ X-α-Gal and 200 ng mL^−1^ AbA, respectively. The positive control was pGBKT7-p53 + pGADT7-T and the negative control was pGBKT7 alone.

### 4.8. Dual-Luciferase Assay

Dual-luciferase assays were performed with tobacco plants (*N. benthamiana*) as described by Chen et al. (2017) [43]. In brief, the ORF sequences of *MaBBX51*, *MaBBX20*, *MaHY5*, and *MaMybA* were cloned into pGreenII 62-SK to generate the effector constructs, whereas the promoter sequences of *MaMybA* and *MaDFR* were fused into pGreenII 0800-Luc to generate the reporter constructs. The primers used are listed in Appendix A. All constructs were individually transformed into *A. tumefaciens* GV3101 cells together with the pSoup vector using the freeze-thaw method. The luciferase-to-Renilla luciferase activity ratio was measured using the Dual Luciferase Reporter Gene Assay Kit (NovoProtein, Shanghai, China) with an Infinite M200 luminometer (Tecan, Männedorf, Switzerland). Statistical analysis was conducted using at least three biological replicates.

### 4.9. Measurement of Total Anthocyanin Content

Anthocyanins were extracted from grape hyacinth corollas under light and shade treatment conditions and from fully opened flower limbs of transgenic tobacco lines as described by Chen et al. (2017) [43]. The experimental conditions, instruments, and protocols of reverse HPLC analysis were as described by Chen et al. (2017) [43]. Anthocyanin content was determined based on a standard curve generated using cyanidin 3-rutinoside. All samples were analysed using three technical and biological replicates.

### 4.10. Statistical Analysis

Data are presented as the means ± standard deviation (SD) and were analysed using the *L**east*
*S**ignificant*
*D**ifference* (*LSD*) test; cut-off, *p* < 0.05. Independent sample t-test and analysis of variance (ANOVA) were performed using the SPSS 20.0 software (SPSS Inc., Chicago, IL, USA).

## Figures and Tables

**Figure 1 ijms-23-05678-f001:**
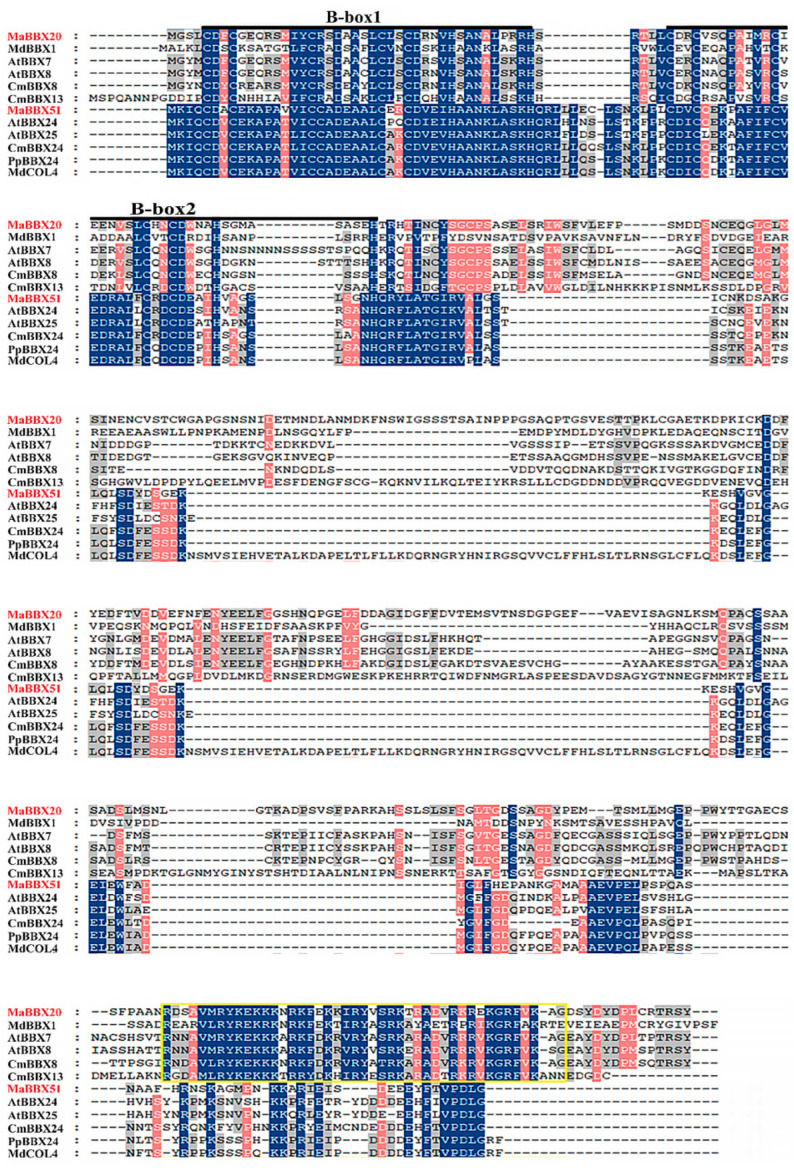
Multiple sequence alignment of MaBBX20, MaBBX51, and other BBXs. The alignment was generated using the GeneDoc software. Blue, pink, and grey columns indicate 100%, 80%, and 60% identity, respectively. The B-box and CCT domains are marked by dark lines and the yellow square, respectively. GeneBank accession numbers for all proteins are listed in Appendix A.

**Figure 2 ijms-23-05678-f002:**
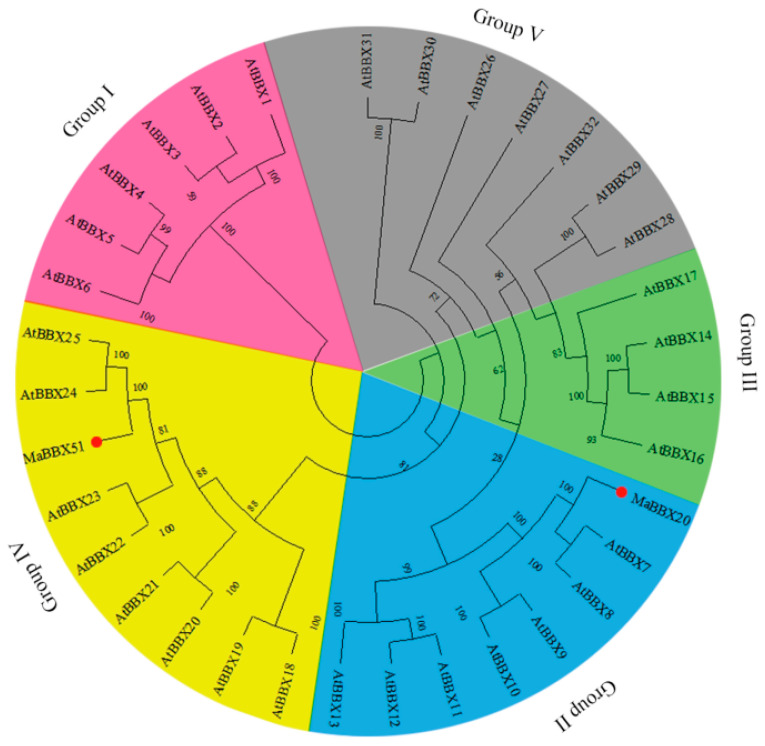
Phylogenetic analysis of MaBBX20, MaBBX51, and 32 *A. thaliana* BBX transcription factors. The neighbour-joining phylogenetic tree was constructed using the MEGAX software. Numbers next to each branch represent the bootstrap values of 1000 replicates. MaBBX20 and MaBBX51 are marked by red dots. Five subfamilies (I to V) are indicated in pink, blue, green, yellow, and grey. GenBank accession numbers for all proteins are listed in Appendix A.

**Figure 3 ijms-23-05678-f003:**
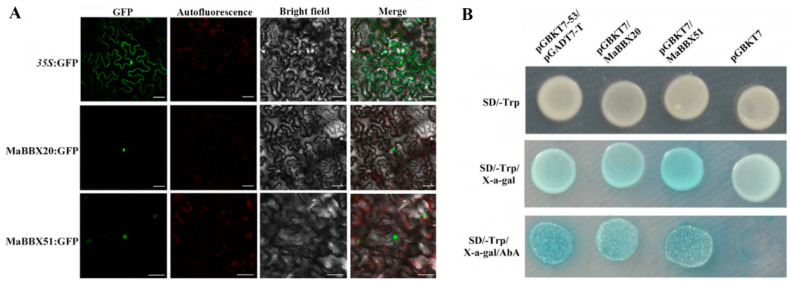
Subcellular localisation and transcription activation ability of MaBBX20 and MaBBX51. (**A**) After transient transformation of *N. benthamiana* leaves with *35S:GFP*, *35S:MaBBX20:GFP*, and *35S:MaBBX51:GFP*, MaBBX20 and MaBBX51 were found localised in the nucleus. (**B**) Transcription activation ability of MaBBX20 and MaBBX51 in the yeast system.

**Figure 4 ijms-23-05678-f004:**
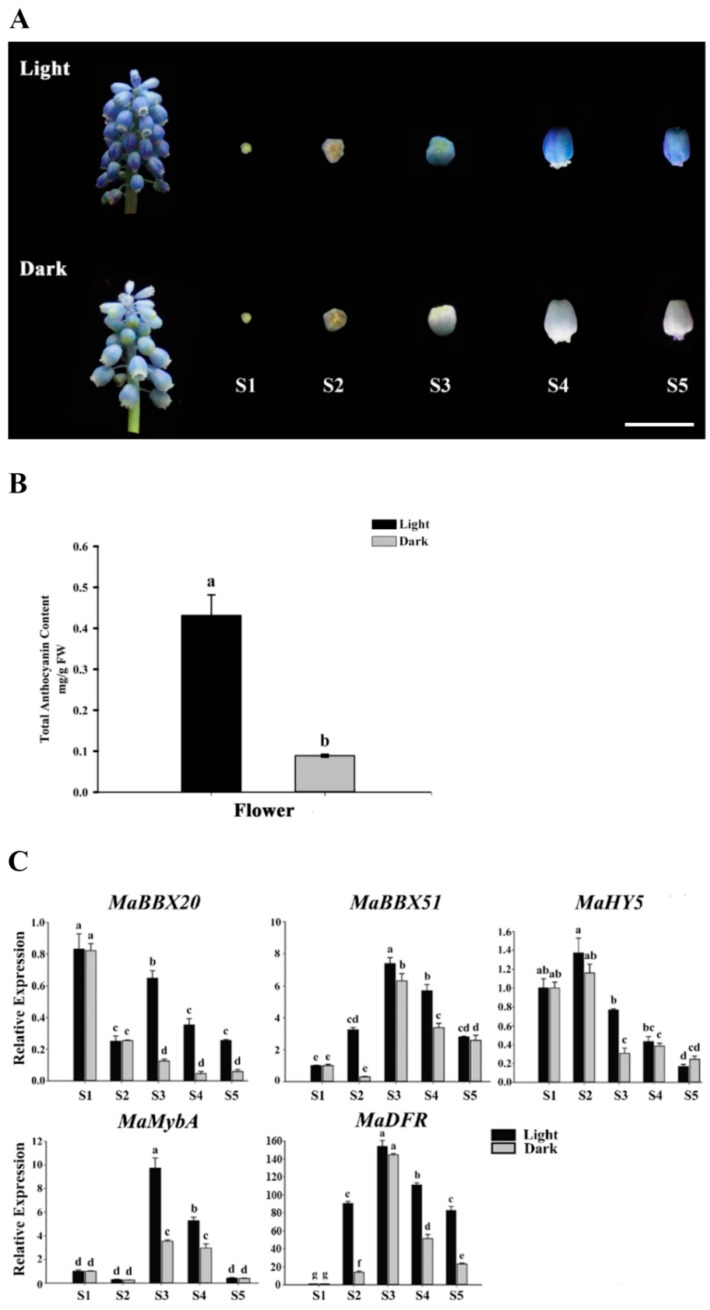
Flower anthocyanin levels and expression patterns of *MaBBX20* and *MaBBX51* under shade treatment in *M. aucheri* ‘Dark Eyes’. (**A**) Shade treatment of inflorescence blocked anthocyanin accumulation. Scale bar: 1 cm. (**B**) Shade treatment reduced the total anthocyanin content of inflorescence. FW, fresh weight. (**C**) qPCR analysis of *MaBBX20*, *MaBBX51*, *MaHY5*, *MaMybA*, and *MaDFR* expression during flower development under light and shade conditions. *MaActin* was used as the reference gene. Data are presented as the mean ± SD. Different letters above the bars indicate significantly different values calculated using one-way ANOVA followed by LSD analysis (*p* < 0.05).

**Figure 5 ijms-23-05678-f005:**
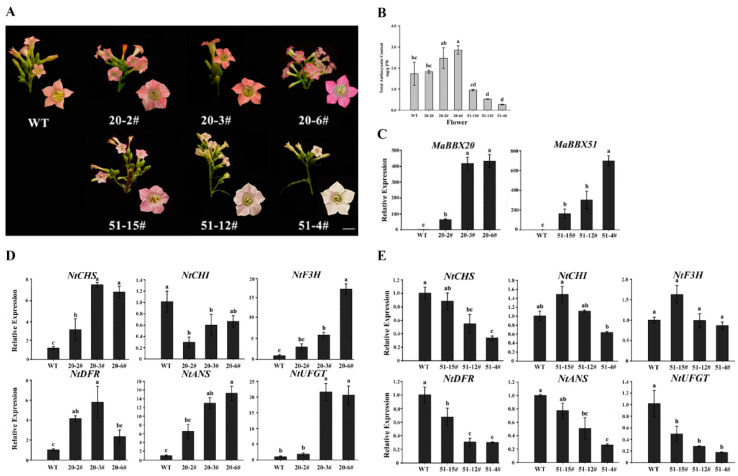
Functional analysis of *MaBBX20* and *MaBBX51* in the tobacco plant. (**A**) Flower phenotypes of overexpressing transgenic lines. Three independent transgenic lines were used for each gene. Scale bar: 2 mm. (**B**) Anthocyanin content in tobacco corollas of transgenic lines. FW, fresh weight. (**C**) qPCR analysis of *MaBBX20* and *MaBBX51* expression in the corollas of their transgenic tobacco. (**D**,**E**) Relative expression levels of *NtCHS*, *NtCHI*, *NtF3H*, *NtDFR*, *NtANS*, and *NtUFGT* in the corollas of transgenic lines overexpressing *MaBBX20* (**D**) and *MaBBX51* (**E**). 20–2#, 20–3#, and 20–6# were transgenic lines of *MaBBX20*; 51–4#, 51–12#, and 51–15# were transgenic lines of *MaBBX5*1. *NtTubA1* was used as an internal control. Data are presented as the mean ± SD. Different letters above the bars indicate significantly different values calculated using one-way ANOVA followed by LSD analysis (*p* < 0.05).

**Figure 6 ijms-23-05678-f006:**
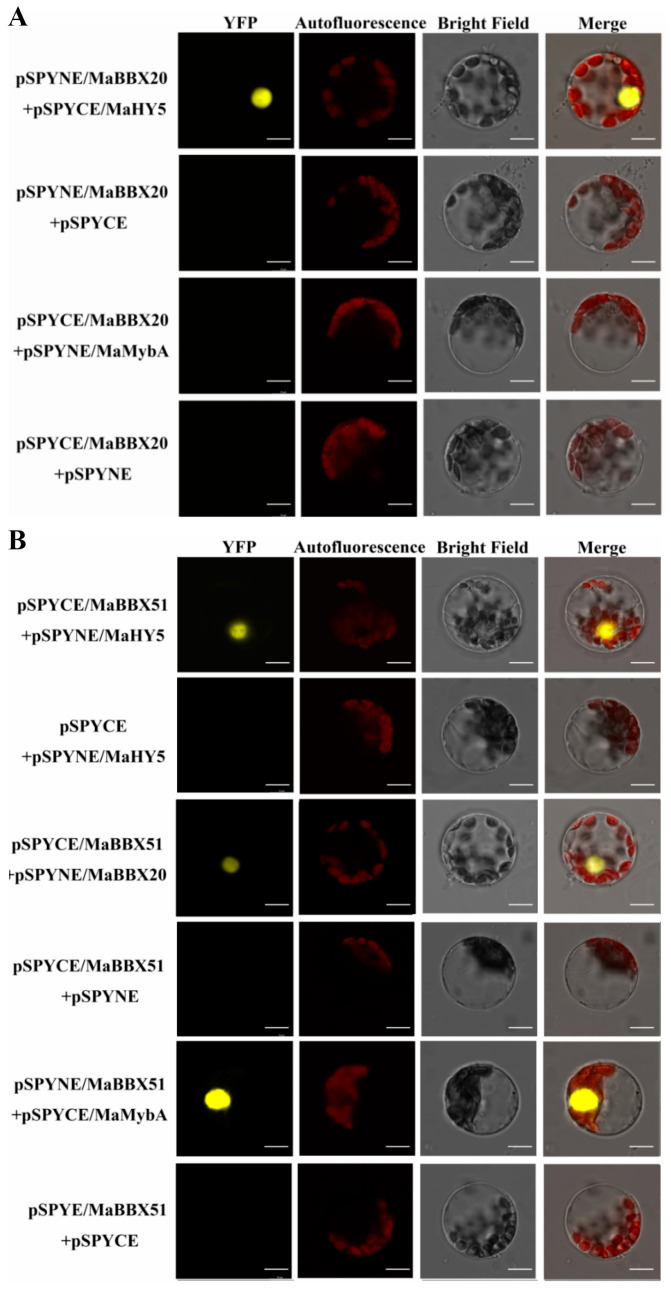
Interaction between MaBBX20, MaBBX51, and other anthocyanin-related proteins in Arabidopsis protoplasts. (**A**) A BiFC assay was carried out to verify the interaction between MaBBX20 and MaHY5 or MaMybA in *Arabidopsis* protoplasts. (**B**) BiFC assay showing MaBBX51 interacting with MaHY5, MaBBX20, and MaMybA. YFP: YFP fluorescence; Autofluorescence: chloroplast autofluorescence; Merge: merged images of chloroplast autofluorescence, YFP fluorescence, and bright-field microscopy. Scale bars: 10 μm.

**Figure 7 ijms-23-05678-f007:**
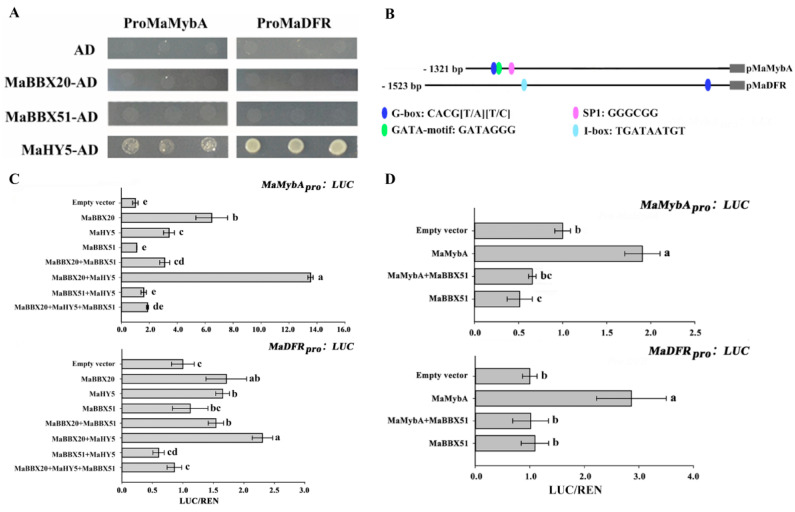
MaBBX20 and MaBBX51, together with MaHY5, coordinate anthocyanin biosynthesis regulation by activating the transcription of MaMybA and MaDFR. (**A**) Yeast one-hybrid assays of MaBBX20, MaBBX51, MaHY5, and the promoters of MaMybA and MaDFR. MaBBX20 and MaBBX51 did not bind to the promoter regions of either MaMybA or MaDFR, whereas MaHY5 did. (**B**) Distribution of light-responsive elements in the promoters of MaMybA and MaDFR. (**C**,**D**) Luciferase/Renilla (LUC/REN) activity ratio to determine the effects of MaBBX20, MaBBX51, and MaHY5 (**C**) MaMybA and MaBBX51 (**D**) on the expression of MaMybA or MaDFR in *N. benthamiana* leaves. Data are presented as the mean ± SD. Different letters above the bars indicate significantly different values calculated using one-way ANOVA followed by LSD analysis (*p* < 0.05).

**Figure 8 ijms-23-05678-f008:**
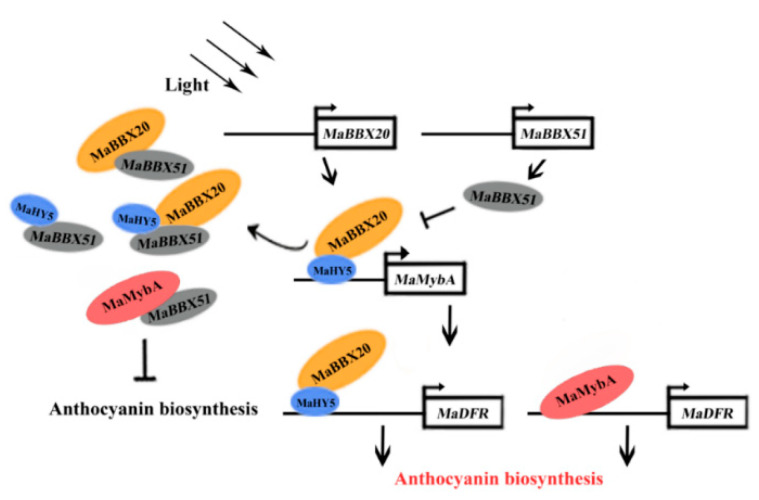
Suggested model for the roles of MaBBX20 and MaBBX51 in light-induced anthocyanin biosynthesis in grape hyacinth. Light activates MaBBX20, MaBBX51, and MaHY5 expression. MaBBX20 then interacts with MaHY5 and forms a MaBBX20–MaHY5 complex to activate MaMybA and MaDFR transcription, thereby positively regulating anthocyanin biosynthesis. Conversely, MaBBX51 interacts with MaHY5 and MaBBX20, hampering the formation of the MaBBX20–MaHY5 complex and subsequently inhibiting MaMybA and MaDFR transcription. MaBBX51 can also interact with MaMybA, inhibiting its binding to its promoter or that of MaDFR, thereby negatively regulating anthocyanin biosynthesis.

## Data Availability

The original contributions presented in the study are included in the article/Appendix A. Further inquiries can be directed to the corresponding author/s.

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
