# Peer review of "Two B-Box Proteins, MaBBX20 and MaBBX51, Coordinate Light-Induced Anthocyanin Biosynthesis in Grape Hyacinth"

_ijms, 2022, doi:10.3390/ijms23105678_

Round 1
Reviewer 1 Report
The present work is interesting as it documents the role of MaBBX20 and MaBBX51 during flower colouration in grape hyacinth (Muscari spp.). However, I have some comments/suggestions for the authors to improve their work:
-Figure 1, the quality is low, please replace
-line 161, delete the spaces between 35S: and the following gene
-line 162, "the green fluorescence signal of the control 35S: GFP was distributed throughout the cell". I do not agree with this sentence because it is visible that the fluorescence is not in the cytoplasm but only in the apoplast. Please, take into consideration this for all the related comments in the MS.
-Figure 4 and 5, the data are expressed as relative expression. Relative to what? No sample is considered 100% or 1 unit. It is not clear.. it seems an absolute quantitation but the related methods also are not clear.
-the caption of figure 6 should better explain what do the colors mean!
-the primer sequences used in PCR analyses were obtained by other works or specifically designed here? In the first case, please, mention the source in the table of supplemental material; in the second specify the method.
-in the introduction and/or in discussion the authors should better underline the fact that, especially for phenolics compounds (including anthocyanins and other types of flavonoids, but also simple phenols) the light stimuli are essential. This issue is already known in literature but the authors do not stress it, while I believe that reporting how other secondary metabolites have been observed to increase under different light, altitude, temperature variation can valorise the present work. I suggest to mention the following papers: Journal of Applied Research on Medicinal and Aromatic Plants, 2015, 2.4: 105-113; Biochemical Systematics and Ecology, 2021, 95: 104231; Molecules, 2018, 23.4: 762; Oecologia, 2009, 160.1: 1-8.
Author Response
We thank the reviewer and editors for your advices. Comments and response for our manuscript entitled “Two B-box Proteins, MaBBX20 and MaBBX51, Coordinate Light-Induced Anthocyanin Biosynthesis in Grape Hyacinth”. Please see the attachment for our response.

Reviewer 2 Report
The manuscript describes investigations of MaBBX20 and MaBBX51 proteins involved in the control of anthocyanin biosynthesis by light in grape hyacinth and compares with similar investigations, mainly in Arabidopsis, published on other transcription factors. The manuscript is well structured, the experimental approach, the interpretation of the results and the conclusions agree with current views in a field where gene and its encoded protein are spoked out without distinction. The difficulty of in vitro transcription assays makes useful the genetic approaches of the manuscript despite the speculative components of the conclusions.
Minor English revision is necessary, and some scientific points should be addressed and corrected.
One table describing transcription-related abbreviations and motifs in DNA would be useful.
In the last paragraph of Introduction, the phrase “we characterized two BBX proteins…” is not correct. The manuscript does not characterize any protein, it deals on (characterizes) the function of proteins encoded by two genes. The technologies used to find regulation by MaBBX20/51 must be briefly named here and in the abstract.
Figure 4B. Why not anthocyanin in S1, S2, S3, S4 and S5? It would be important to compare with MaBBX20 and MaBBX51 evolution during flower development.
Lines 231-232, “Compared with the WT control line, total anthocyanin levels of MaBBX20 OE lines were slightly increased, whereas those of MaBBX51 OE lines were significantly reduced (Figure 5B).” However, the comparison of Figures 2B and 5B suggests that the high level of anthocyanin in tobacco corollas would mask changes in transgenics and decrease the relevance of the assays shown in Figure 5B.
Figures 5B,C,D,E. What numbers in X-axes mean?
Author Response

(The authors gave the same response as above.)
